# Nisin as a Novel Feed Additive: The Effects on Gut Microbial Modulation and Activity, Histological Parameters, and Growth Performance of Broiler Chickens

**DOI:** 10.3390/ani10010101

**Published:** 2020-01-08

**Authors:** Bartosz Kierończyk, Mateusz Rawski, Zuzanna Mikołajczak, Sylwester Świątkiewicz, Damian Józefiak

**Affiliations:** 1Department of Animal Nutrition, Poznań University of Life Sciences, Wołyńska 33, 60-637 Poznań, Poland; zuzanna.mikolajczak@up.poznan.pl (Z.M.); damian.jozefiak@up.poznan.pl (D.J.); 2Division of Inland Fisheries and Aquaculture, Institute of Zoology, Poznań University of Life Sciences, Wojska Polskiego 71c, 60-625 Poznań, Poland; mateusz.rawski@up.poznan.pl; 3Department of Animal Nutrition Physiology, National Research Institute of Animal Production, 32-083 Balice, Poland; s.swiatkiewicz@izoo.krakow.pl

**Keywords:** bacteriocins, nisin, E234, monensin, feed additive, broiler chicken, performance, microbiota, bacterial fermentation, histomorphology

## Abstract

**Simple Summary:**

In an era with an increasing number of antibiotic-resistant bacteria strains, there is a need to find a novel and efficient alternative to the antibiotics commonly used in animal nutrition. As natural proteins synthesized by most known bacteria, bacteriocins are considered future candidates. To date, nisin (E234), the best-known bacteriocin, is used as a preservative against food-borne pathogens such as *Listeria monocytogenes* in the food industry. However, there are scarce data about the application of nisin in the diet of livestock, including poultry. In this study, we evaluated the effect of nisin in broiler chicken diets on selected microbial populations, the activities of which are related to gastrointestinal health, growth performance, and gut histomorphology. We found that nisin application positively affects the feed conversion ratio and exerts a similar effect as the ionophore coccidiostat monensin in the case of microbiota modulation. Additionally, nisin supplementation decreased microbial fermentation in the jejunum. No changes in ileal histomorphology or internal organ weights were noted. We conclude that nisin may be considered a natural and safe antimicrobial agent and growth promoter in broiler chicken nutrition.

**Abstract:**

Two independent experiments were performed to evaluate the effect of nisin alone or with monensin on gut microbiota, gut microbial activities, and histomorphology (exp 1) and the effect of nisin application in a dose‒response manner on the growth performance of broiler chickens (exp 2). A total of 900 one-day-old female Ross 308 chicks (400, exp 1; 500, exp 2) were randomly distributed to four groups (exp 1; 10 replicate pens per treatment with 10 birds each), i.e., NA, no additives; MON, monensin (100 ppm); NIS, nisin (2700 IU/kg diet); and MON + NIS, a mixture of monensin (100 ppm) and nisin (2700 IU/kg diet); or 5 treatments (exp 2), NA, no additives; NIS_100_, nisin (100 IU/kg diet); NIS_200_, nisin (200 IU/kg diet); NIS_400_, nisin (400 IU/kg diet); and NIS_800_, nisin (800 IU/kg diet). Nisin supplementation positively affected the microbiota of the gut by reducing potentially pathogenic bacterial populations in the jejunum and ceca. The bacterial fermentation in the jejunum was significantly lowered by nisin addition. The addition of nisin from 100 IU to 800 IU decreased the FCR value over the entire experimental period. According to the results, nisin can be considered a natural dietary supplement for broiler chickens.

## 1. Introduction

Nisin (E234) is one of the most examined bacteriocins used as an inhibitory agent of potentially pathogenic bacteria that proliferate in the food industry [1,2,3]. Bacteriocins are relatively small antimicrobial peptides ribosomally synthesized by most known bacteria, e.g., Firmicutes, Bacteroidetes, Actinobacteria, and Proteobacteria [4]. These substances have activity against both Gram-positive and Gram-negative bacteria, depending on the producing strain [5]. Thus, bacteriocins may be considered novel alternatives for antibiotics and innovative cures in human medicine, as well as veterinary science [6,7]. Nisin (3.5 kDa) is produced by some *Lactococcus lactis* subsp. *lactis* strains and, as a protein compound, comprises 34 amino acid residues [8]. Its antimicrobial properties consist of permeabilization of the bacterial cell membrane by pore formation, leading to leakage of cellular content and disruption of the cell, preventing growth [9]. The U.S. Food and Drug Administration notes that nisin is ‘generally recognized as safe’ (GRAS status; 53 FR 11247), and the European Union approved the usage of nisin as a food additive—E234 (83/463/EEC) [10,11]. The European Food Safety Authority (EFSA) claimed that nisin is digested by protolithic enzymes such as pepsin and trypsin, while other authors mentioned that chymotrypsin, pancreatin, and carboxypeptidases in the gastrointestinal tract (GIT) environment can degrade nisin [12,13]. Therefore, in the literature, it is often assumed that the antibacterial function of this compound is fully eliminated because of protolithic enzymes, and that the compound does not have any impact on the GIT microbiome [14]. In contrast, it is well documented that the use of nisin encapsulated with lipid or bacteriocin without any protection layer may improve the growth of broiler chickens and cause positive GIT microbiota modulations at similar levels [15,16]. In addition, the ability of nisin to change the microbiota was also observed in mice, rats, rabbits, and ruminants [17,18,19,20]. This is in agreement with the results of an in vitro study that showed that partially digested nisin may maintain its activity against pathogenic bacteria; however, the length of the peptide chain has a crucial effect on bacterial growth inhibition [21,22]. Furthermore, according to the available literature, data about the addition of effective levels of bacteriocins, including nisin, as a feed additive on the productivity parameters and microbial changes in the GIT, with an emphasis on pathogen inhibition, are lacking. To date, only Józefiak et al. [23] have presented the influence of various nisin level inclusions, i.e., 100, 300, 900, and 2700 IU of nisin/g, supplemented in broiler chicken diets. Otherwise, only a few in vivo experiments have been carried out to evaluate the effect of nisin on poultry productivity [15,16,24,25]. Therefore, it should be emphasized that the EFSA recommends the continuation of nisin evaluation as an antimicrobial drug for human or livestock use [26]. Therefore, two independent experiments were carried out on broiler chickens to evaluate the effect of dietary nisin alone or in combination with ionophore coccidiostat monensin on the GIT microbiota, microbial activities, and histomorphology (exp 1). The aim of the second experiment was to investigate the effect of relatively small amounts of nisin on growth performance in a dose‒response manner.

## 2. Materials and Methods

### 2.1. Birds and Housing

All procedures and experiments complied with the guidelines and were approved by the Local Ethics Commission for Experiments on Animals of Poznań University of Life Sciences (Poznań, Poland; no 8/2015) with respect to animal experimentation and care of the animals under study, and all efforts were made to minimize suffering.

Two independent experiments were conducted. The first experiment was a continuation of a previous trial described in Kierończyk et al. [15], in which nisin and monensin were added to broiler chicken diets and the effect on growth performance parameters, coefficients of apparent digestibility of nutrients, tibiotarsus chemical composition, and length and weight of selected sections of the gastrointestinal tract was analyzed. In the present work, the effect of nisin and monensin on selected microbial populations and their activity by analyzing the short-chain fatty acids (SCFAs) and pH of digesta was examined. In total, 400 one-day-old female Ross 308 chicks were used and allotted to four groups, with 10 replications (10 birds each). At the end of the first trial (35 days), the birds were slaughtered, eviscerated, and the digesta from the crop, jejunum, and ceca was collected. All details are given in further sections.

The second trial was carried out to investigate the effect of nisin addition in a dose‒response manner, i.e., 100 IU, 200 IU, 400 IU, and 800 IU per kg of broiler chicken diet. A total of 500 one-day-old female Ross 308 chicks were randomly distributed to five dietary treatments, with 10 replicate pens per group and 10 birds per pen. The experiment lasted 35 days, the growth performance parameters, i.e., body weight and feed intake (FI), were determined, and the body weight gain (BWG) and feed conversion ratio (FCR) was calculated. The abovementioned variables were obtained at 14 and 35 days of age using analytical scale (NVL5101, OHAUS, Greifensee, Switzerland) with accuracy ±1 g. The housing conditions were the same in both experiments: the birds were kept in floor pens (1.00 × 1.00 m; straw litter) over 35 days; stock density was established at 10 birds per square meter. Additionally, 9000 birds of the same origin (hatchery) were used to imitate the commercial environment production conditions. The closed chicken house was equipped with an artificial light program (fluorescent lights), automatic heaters, and forced ventilation. The birds were given 23 h of light and 1 h of dark for the first week, followed by 19 h of light and 5 h without from 7 to 21 days of age. From 22–35 d of age, the lighting system was similar to that of the first week. The light intensity was set at 20 lx according to the EU directive (2007/43/EC). The temperature inside the building was set up at 32–33 °C at the beginning of the test and was reduced by 2–3 °C each week. On the 28th day, the temperature was set at 21 °C and was approximately 18 °C at the end of the experiment. The humidity level was in the range of 50–60%. During the experimentation, the maximum concentration of CO_2_, as well as NH_3_ did not exceed 3000 ppm, and 10 ppm, respectively. The rearing conditions were set up accordingly to AVIAGEN guidelines.

### 2.2. Diets and Feeding Program

The composition of the experimental basal diets is shown in Table 1. In both trials, the birds were fed ad libitum and had permanent access to drinking water for 35 d. For each feeding period, all diets were calculated to meet or exceed the nutrient requirements recommended by the NRC [27] for broiler chickens. The experimental diets were designed to provoke GIT colonization by *Clostridium perfringens* via the usage of viscous cereals (wheat and rye), animal dietary fat (pig lard), and fish meal as ingredients [28,29,30]. The diets were prepared in mash form; all of the raw materials were ground by a disc mill (Skiold A/S, Sæby, Denmark) at 2.5 mm disc distance and mixed with no heat treatment. The diets were produced in the Piast Pasze feed mill (Lewkowiec, Poland) according to the ISO 9001:2008 procedures. The feed was prepared on a laboratory-scale line equipped with a horizontal double band mixer (Zuptor, Gostyń, Poland) with roller mills (Skiold, Sæby, Denmark). Initial diets were offered to all birds from 1 to 14 d of age, and grower‒finisher diets were offered from 15 to 35 d of age. No exogenous enzymes were used in the studies. The design of experiment 1 was as follows: NA, control diet with no additives; MON, monensin addition (100 ppm); NIS, nisin preparation (2700 IU/kg diet); and MON + NIS, a mixture of monensin (100 ppm) and nisin (2700 IU/kg diet). In the second trial, no ionophore coccidiostat was used, and the following treatments were applied: NA, control diet with no additives; NIS_100_, diet with the addition of nisin preparation (100 IU/kg diet); NIS_200_, nisin supplementation (200 IU/kg diet); NIS_400_, nisin preparation (400 IU/kg diet); and NIS_800_, nisin addition (800 IU/kg diet).

### 2.3. Preparation of Nisin

Nisin was prepared according to the method elaborated by the Department of Biotechnology and Food Microbiology, Poznań University of Life Sciences, using *Lactococcus lactis* subsp. *lactis* (ATCC11454). All details regarding the preparation and concentration analyses of nisin were reported previously by Józefiak et al. [23] and Kierończyk et al. [15].

### 2.4. Data and Sample Collection

At the end of the first experiment (d 35), 10 randomly chosen birds (one broiler chicken from each of 10 replicate pens per treatment) were sacrificed by cervical dislocation to collect material for further analyses. The pH values of the digesta samples from the crop, jejunum, and ceca were measured immediately after slaughter using a pH meter with combined glass and reference electrode (VWR International, pH 1000 L, Leuven, Belgium). The portion of the jejunal samples was gently squeezed, immediately packed, sealed in sterilized plastic bags, frozen, and stored at −80 °C for analysis of the selected microbial populations by fluorescent in situ hybridization (FISH) of single bacterial cells and organic acids. The jejunum was considered to begin at the end of the duodenum and end at Meckel’s diverticulum. The ileum was defined as the small intestinal segment caudal to Meckel’s diverticulum. Additionally, the ileal tissue (1 cm after Meckel’s diverticulum) was collected after evisceration for histomorphology analysis. The selected internal organ weights in relation to body weight (BW; % BW), i.e., Bursa of Fabricius, spleen, liver, and pancreas, were measured. After slaughter, the abovementioned organs were rinsed in sterile water, drained, weighed using the electronic balance PS 600/C/2 (Radwag, Radom, Poland) precision scales and measured. The measurement was made with accuracy to 3 decimals. In the first experiment, the growth performance parameters, i.e., BWG, FI, and FCR, were analyzed on days 14 and 35.

### 2.5. Analysis of the Microbial Community and Its Activity

All details of sample preparation and FISH analyses for bacteria enumeration from jejunal digesta were described by Rawski et al. [31], and Józefiak et al. [23]. The oligonucleotide probes used in this study are presented in Table 2. The concentration of organic acids in the digesta of the various GIT locations was determined by gas chromatography (Model 6890, Hewlett Packard, Agilent Technologies, Nærum, Denmark) according to Canibe et al. [32].

### 2.6. Histological Analyses

The histological analysis of ileal samples was performed according to Rawski et al. [31]. Briefly, the ileal samples were fixed immediately in freshly prepared formaldehyde solution (40 g/L of formaldehyde prepared in 0.01 M PBS, pH = 7.4) and incubated 12 h. Ileal fragments were dehydrated in alcohol dilutions, stowed in xylene, and embedded in paraffin. At the least 10 slides (5 μm) per block were stained using haematoxylin and eosin. The obtained material was analyzed under a light microscope (Axiophot, Carl Zeiss, Germany) with 5 × 5 magnification. The length of villi was measured from the top of the epithelium villi to the junction with the crypt. In the cross-sections, the lengths of all villi with a complete structure were measured. Destroyed villi were excluded from the trial. Mucosal thickness was determined as the distance between the mucosal epithelium and the muscular layer, and the muscularis was determined as the inner circular and outer longitudinal layers of smooth muscle cells [38]. The measurements were made on 10 serial slides using a micrometer glass master (0.01 mm, PZO, Warsaw, Poland) and treated as the means.

### 2.7. Statistical Analysis

The experiments had a completely randomized design. All data were tested for normal distributions using the Kolmogorov‒Smirnov test. An analysis of variance was conducted using Bartlett’s test. The significance of differences among groups was determined with Duncan’s multiple range test at the significance level of *p* < 0.05. The analyses were performed using SAS software (SAS Institute Inc., Cary, NC, USA).

In the first trial, two factorial designs were applied according to the following general model:Y_ij_ = μ + α_i_ + β_j_ + (αβ)_ij_ + δ_ij_,
where Y_ij_ was the observed dependent variable, μ was the overall mean, α_i_ was the effect of monensin, β_j_ was the effect nisin, (αβ)_ij_ was the interaction between monensin and nisin, and δ_ij_ was the random error.

In the second experiment, the following general model was used:Y_i_ = μ + α_i_ + δ_ij_,
where Y_i_ is the observed dependent variable, μ is the overall mean, α_i_ is the effect of nisin, and δ_ij_ is the random error.

## 3. Results

### 3.1. Experiment 1

The effect of nisin and monensin on the pH values of digesta in selected GIT segments is presented in Table 3. No influence of ionophore coccidiostat or bacteriocin addition on the crop digesta pH was observed (*p* > 0.05). However, the addition of both nisin (*p* < 0.001) and monensin (*p* = 0.003) increased the pH of jejunal content. The opposite, i.e., decreased pH values in the ceca, was observed after nisin (*p* = 0.039) and monensin (*p* = 0.002) supplementation.

The selected microbial populations in the jejunal digesta are presented in Table 4. Interactions among the experimental factors were observed for the total number of bacteria (DAPI; *p* < 0.001), Enterobacteriaceae (*p* < 0.001), *Clostridium perfringens* (*p* < 0.001), and *Lactobacillus* sp./*Enterococcus* sp. (*p* < 0.001). Addition of monensin or nisin separately reduced the abovementioned groups of bacteria compared to the NA control treatment. However, the highest effect on the concentration of the abovementioned bacteria was observed in the NIS treatment (*p* < 0.05). Only the concentration of *Lactobacillus* sp./*Enterococcus* sp. was not affected by the mixture of experimental factors compared to NA. Furthermore, nisin supplementation significantly decreased the number of the *Clostridium leptum* subgroup (*p* < 0.001) and the *Clostridium coccoides*-*Eubacterium rectale* cluster (*p* < 0.001). The main effect of monensin was the reduction in the *Clostridium coccoides*-*Eubacterium rectale* cluster (*p* = 0.002). The experimental factors had not effect on the *Bacteroides*-*Prevotella* cluster (*p* = 0.213).

The cecal microbial community is shown in Table 5. Nisin addition to the broiler chicken diets decreased Enterobacteriaceae (*p* < 0.001), the *Bacteroides*-*Prevotella* cluster (*p* < 0.001), *Clostridium perfringens* (*p* < 0.001), *Lactobacillus* sp./*Enterococcus* sp. (*p* < 0.001), the *Clostridium leptum* subgroup (*p* < 0.001), as well as the *Clostridium coccoides*-*Eubacterium rectale* cluster (*p* < 0.001), in comparison to the control group (NA). Similar to nisin, monensin also affected (*p* < 0.05) these selected microbial populations. However, no additive or synergistic effect of experimental factors was observed. Interactions between nisin and monensin were observed except for DAPI (*p* = 0.468). However, both exhibited the greatest effect on the increased (*p* < 0.05) total bacteria count.

The results of the short-chain fatty acid measurements are shown in Table 6 and Table 7. In the case of microbial fermentation in the jejunum (Table 7), nisin application lowered the sum of volatile fatty acids (VFA) (*p* = 0.007), mainly via reducing the acetic acid concentration (*p* = 0.024). There was no effect on the microbial fermentation in the jejunum after monensin application (*p* > 0.05). The opposite effect was noticed in the ceca, where nisin application enhanced (*p* = 0.050) the concentration of acetic acid. The tendency (*p* = 0.058) of increasing VFA was also observed after nisin supplementation. Additionally, monensin addition increased (*p* = 0.030) the butyric acid, as well as iso-valeric acid (*p* = 0.050) content without affecting the final the sum of VFA (*p* = 0.177) in ceca. There was no interaction (*p* > 0.05) between the experimental factors.

Table 8 presents the results of the ileal histomorphometry measurements. No significant interaction (*p* > 0.05), as well as effects of experimental factors (*p* > 0.05) were noticed in the case of villus high, crypt depth, their ratio, mucosa, and muscular layer thickness.

### 3.2. Experiment 2

The growth performance results are shown in Table 9. In general, the addition of nisin ranging from 100 IU to 800 IU per kg of broiler chicken diet did not significantly affect the BWG or FI (*p* > 0.05). However, the addition of the following activities of nisin, i.e., 100 IU, 200 IU, and 800 IU, decreased the FCR values compared to the control group (NA) at 14–35 d (*p* < 0.001). In terms of the entire experimental period, all proposed activities of bacteriocin lowered the FCR value (*p* < 0.001). In the first two weeks, there was no significant effect of nisin supplementation on the FCR (*p* = 0.159). There were no significant differences between groups (*p* > 0.05) in terms of the selected internal organ weights (Table 10).

## 4. Discussion

According to the EFSA report, nisin is considered a novel antimicrobial drug for humans as well as domestic animals, and it is recommended that its mode of action under in vivo conditions be further evaluated [26]. Hitherto, there are a few commercially available products containing nisin that are used to prevent and cure mastitis [39,40]. Furthermore, nisin has been widely examined against *Staphylococcus aureus*-induced skin infections, dental caries, and apoptosis of cancer cells factor [41,42,43]. Nisin usage as a food preservative against mainly *Listeria monocytogenes* is thought of as safe because it is degraded by endogenous proteolytic enzymes in the GIT [44,45]. In human nutrition, the average daily intake (ADI) of nisin was updated from 1 mg per kg of body weight (BW) to 12 mg/kg BW (unripened cheese) and 25 mg/kg BW (heat-treated meat products) [26]. In contrast, bacteriocins are forbidden for use in livestock diets, including poultry, and bacteriocins are not registered as feed additives (EU 1831/2003). Surprisingly, nisin is able to maintain its antimicrobial activity after digestion depending on the resulting fragments [21]. This is in agreement with the results of Józefiak et al. [23], where significant changes in the GIT microbiota were noticed by the positive reduction in potentially pathogenic groups of bacteria, i.e., the *Bacteroides*-*Prevotella* cluster, and Enterobacteriaceae in the ileum after nisin supplementation. Additionally, Kierończyk et al. [16] highlighted the positive effect of nisin on the reduction in the proliferation of *Clostridium perfringens* and *Lactobacillus* sp./*Enterococcus* sp. in this segment. The results of the present study confirmed the antimicrobial properties of nisin in both the jejunum and ceca. In addition to previously mentioned microbial populations, nisin has limited the number of *Clostridium leptum* subgroup, and the *Clostridium coccoides*-*Eubacterium rectale* cluster. The positive effects of nisin application in broiler chicken diets on changes in the microbiota consist not only of a reduction in pathogen occurrence in the chicken GIT, but also of lowering the competition for nutrients between bacteria and the host, improving energy utilization by decreasing the number of bacteria from the genera *Lactobacillus*, *Clostridium*, and *Bacteroides* [46,47]. The activity measurements, i.e., pH and SCFA concentration, were consistent with the microbiology results. The increasing pH value in the jejunum is a result of the reduction in *Lactobacillus* sp./*Enterococcus* sp. population and low acetic acid fermentation, as well as the sum of the VFA. This is in agreement with the results of Józefiak et al. [23], who found that nisin (2700 IU) significantly decreased the total SCFA concentration in the ileum. However, in the present study, the cecum fermentation tended (*p* = 0.058) to increase with increasing acetic acid concentration after nisin addition to the chicken diet, while other authors observed contradicting results. Nevertheless, the lower activity of nisin, i.e., 100 IU, 300 IU, 900 IU, did not affect the cecum fermentation [23]. In the present study, the effects of nisin on the microbiota fermentation (ileal and cecal) may be explained by its main antimicrobial targeting. It is well documented that bacteriocins inhibit the growth and development of bacteria especially in the case of closely related taxa, i.e., across genera or the same species [48]. Due to this fact, the nisin produced by the *L. lactis* subsp. *lactis* may have the main impact on lactic acid bacteria (LAB), thus the microbial fermentation is reduced in the higher GIT segments where they occur as dominant, i.e., the crop (10^9^ cells g^−1^), gizzard (10^8^ cells g^−1^) as well as small intestine (10^9^–10^11^ g cells g^−1^) [49,50,51]. Contrary to the ceca where the LAB populations (*Bacillus*-*Lactobacillus*-*Streptococcus* subdivision) are in minority [52]. The abovementioned mechanism seems to be confirmed by the fact that the fermentation in ceca was not reduced as much as in the upper parts due to the presence of wide spectra of bacterial populations which could be resistant to nisin activity. Additionally, the effect of monensin was noticed by the positive reduction of iso-valeric acid concentration, which is a component of putrefactive SCFA (PSFA). It is well known that PSCFA is related to protein fermentation in the ceca by, e.g., *Clostridium perfringens*, and Enterobacteriaceae [53,54]. It is in agreement with obtained results, where the proliferation of abovementioned bacteria was inhibited. Moreover, the decreasing number of volatile fatty acid-producing strains in the ceca result in increased concentrations of propionic, acetic, as well as butyric acid [55]. In the current study, only butyric acid fermentation has been enhanced by monensin addition, while the *Clostridium leptum* subgroup and *Clostridium coccoides*-*Eubacterium rectale* cluster were lowered by the coccidiostat. Nevertheless, the increased level of their activity may have a beneficial impact on the GIT microbiota populations [56]. It should be emphasized that nisin exerts a similar mode of action to salinomycin in terms of antimicrobial properties, as well as the growth performance parameters [16,23]. As the present results have shown, the ionophore coccidiostat monensin also has convergent activity with nisin in the case of microbiota modulation. However, no additive or synergistic effect was observed in the selected microbial populations.

There is still a lack of data on the inclusion level or activity of bacteriocins, including nisin, which may be efficiently used as antimicrobial agents, as well as growth promoters in poultry nutrition. In terms of broiler chickens, only two bacteriocins, i.e., divercin AS7 and nisin, have been partially examined in terms of their dosage [23,57]. Hitherto, it was mentioned that nisin at high activity (900 IU and 2700 IU) has shown the most efficient performance in broilers. In general, supplementation of nisin to chicken diets causes increased the BWG and FI, simultaneously decreasing the FCR [23]. However, it should be mentioned that these results are particularly marked in the first rearing period, i.e., until 14 d of age. This is in agreement with Kierończyk et al. [16], where the positive effect of nisin on the BWG, FI, and FCR was noticed only in the first period. In contrast to the above, divercin AS7 characterized by even low activity, i.e., 100 AU, 200 AU, and 300 AU, positively affected the FCR value of broiler chickens [57]. Additionally, divercin AS7 exerted a similar effect on the BWG and FCR as salinomycin [58]. In the present experiment, the growth performance results revealed a positive role of nisin as a growth promoter. Each dosage of bacteriocin significantly reduced (*p* < 0.001) the FCR value in the last period of the experiment (14–35 d) and throughout the entire trial (1–35 d). Although monensin was not examined as a growth promoter in this study, in the available literature the positive effect of nisin on growth performance results was comparable to that of ionophore coccidiostats, i.e., salinomycin, as well as monensin [15].

Referring to the results of Kierończyk et al. [15], which showed no significant changes in apparent ileal digestibility of crude protein and ether extract after nisin addition, the present study confirmed no detrimental influence on the villus high, crypt depth, mucosa, and muscular layer thickness. The reduction in the length and weight of the duodenum, jejunum, ileum, and cecum have been noted [15]. To date, the histomorphology of the GIT after nisin application was measured only in rabbits, and no effect was observed on the villus surface, height, and crypt depth. Additionally, the application of albusin B, as well as bacteriocin B602, to poultry diets confirmed that these parameters were not affected [59,60].

## 5. Conclusions

The results of the current experiments emphasized that nisin addition to broiler chicken diets may be considered a novel and natural growth promoter that improves feed utilization even at low levels. Moreover, it was confirmed that nisin plays a significant role in the positive modulation of the microbiota of the GIT of broiler chickens via the reduction in the proliferation and populations of potentially pathogenic bacteria, which may negatively alter nutrient utilization. A limitation of bacterial activity was observed in the jejunal digesta, while cecal fermentation tended to be enhanced. Additionally, a positive interaction between bacteriocin nisin and ionophore coccidiostat‒monensin was observed without any additive or synergetic effect. The findings of the present study suggest that nisin exerts a mode of action similar to that of monensin in the scope of antimicrobial properties. Additional studies in terms of the evaluation of anticoccidial properties of nisin are recommended.

## Figures and Tables

**Table 1 animals-10-00101-t001:** Composition and nutritive value of the basal diets, experiments 1 and 2.

Ingredient, g·kg^−1^	Diets	
	1–14 d	15–35 d
Wheat	468.7	487.5
Rye	100.0	100.0
Rapeseed meal 34.0%	100.0	100.0
Soybean meal 46.8%	222.2	186.8
Fish meal 64%	20.0	20.0
Pig lard	55.7	79.8
Vitamin-mineral premix ^1^	3.0	3.0
Dicalcium phosphate	19.5	12.5
Limestone	1.0	1.6
NaCl	1.4	1.6
Na_2_CO_3_	1.5	1.0
L-Lysine	2.4	2.1
DL-Methionine	3.2	2.6
L-Threonine	1.4	1.5
Calculated nutritive value, g·kg^−1^		
AME_N_ (MJ/kg) ^2^	12.3	13.3
Crude protein	215.0	200.0
Crude fat	71.0	94.8
Crude fiber	33.3	32.3
Calcium	8.5	7.0
Lysine	12.5	11.3
Methionine	6.1	5.4
Methionine + cystine	3.8	3.6
Threonine	9.9	9.0

^1^ Provided the following per kilogram of diet: vitamin A, 11.166 IU; cholecalciferol, 2.500 IU; vitamin E, 80 mg; menadione, 2.50 mg; B12, 0.02 mg; folic acid, 1.17 mg; choline, 379 mg; d-pantothenic acid, 12.50 mg; riboflavin, 7.0 mg; niacin, 41.67 mg; thiamine, 2.17 mg; d-biotin, 0.18 mg; pyridoxine, 4.0 mg; ethoxyquin, 0.09 mg; Mn (MnO_2_), 73 mg; Zn (ZnO), 55 mg; Fe (FeSO_4_), 45 mg; Cu (CuSO_4_), 20 mg; I (CaI_2_O_6_), 0.62 mg; Se (Na_2_SeO_3_), 0.3 mg. ^2^ Apparent metabolizable energy corrected to zero nitrogen balance.

**Table 2 animals-10-00101-t002:** Oligonucleotide probes.

Target	Probe	Sequence (5′ to 3′)	References
Enterobacteriaceae	Enter1432	CTT TTG CAA CCC ACT	[33]
*Bacteroides*-*Prevotella* cluster	Bac303	CCAATGTGGGGGACCTT	[34]
*Clostridium leptum* subgroup	Clept1240	GTTTTRTCAACGGCAGTC	[33]
*Clostridium coccoides**-Eubacterium rectale* cluster	Erec482	GCTTCTTAGTCARGTACCG	[35]
*Clostridium perfringens*	Cperf191	GTAGTAAGTTGGTTTCCTCG	[36]
*Lactobacillus* sp./*Enterococcus* sp.	Lab158	GGTATTAGCAYCTGTTTCCA	[37]

**Table 3 animals-10-00101-t003:** The effect of dietary supplementation of nisin alone or in combination with monensin on the pH value of the crop, jejunal, and cecal content, Experiment 1.

Item	Treatments						Main Effects				*p*-Value		
	NA ^1^	MON ^2^	NIS ^3^	MON + NIS ^4^	RMSE ^5^	MON		NIS		Effect of Treatments		Interaction
						−	+	−	+	MON	NIS	MON × NIS
Crop	5.10	5.27	4.97	5.08	0.29	5.04	5.18	5.19	5.03	0.322	0.248	0.850
Jejunum	5.77	6.43	6.59	6.91	0.37	6.20 ^b^	6.67 ^a^	6.11 ^b^	6.75 ^a^	0.003	<0.001	0.289
Ceca	6.45	6.25	6.39	5.79	0.22	6.41 ^a^	6.02 ^b^	6.35 ^a^	6.09 ^b^	0.002	0.039	0.105

^a^^,b^ Means not sharing a common superscript differ significantly (*p* < 0.05); ^1^ control diet with no additives; ^2^ monensin addition (100 ppm); ^3^ nisin preparation (2700 IU/kg diet); ^4^ a mixture of monensin (100 ppm) and nisin (2700 IU/kg diet); ^5^ root-mean-square error; means represent 10 pens of one chick each (10 replicates).

**Table 4 animals-10-00101-t004:** The effect of dietary supplementation of nisin alone or in combination with monensin on selected microbial populations (log CFU/g digesta) in the jejunal content determined by DAPI staining and fluorescent in situ hybridization (FISH), Experiment 1.

Item	Treatments				RMSE ^5^	Main Effects				*p*-Value		
	NA ^1^	MON ^2^	NIS ^3^	MON + NIS ^4^		MON		NIS		Effect of Treatments		Interaction
						−	+	−	+	MON	NIS	MON × NIS
DAPI	9.65 ^a^	9.39 ^b^	9.28 ^c^	9.27 ^c^	0.16	9.46 ^a^	9.33 ^b^	9.52 ^a^	9.28 ^b^	<0.001	<0.001	<0.001
Enterobacteriaceae	8.78 ^a^	8.52 ^b^	8.29 ^c^	8.35 ^c^	0.16	8.53 ^a^	8.43 ^b^	8.65 ^a^	8.32 ^b^	0.007	<0.001	<0.001
*Bacteroides*-*Prevotella* cluster	8.86	8.84	8.79	8.86	0.17	8.82	8.85	8.85	8.82	0.464	0.391	0.213
*Clostridium perfringens*	8.87 ^a^	8.68 ^b^	8.46 ^c^	8.58 ^b^	0.18	8.66	8.63	8.77 ^a^	8.52 ^b^	0.405	<0.001	<0.001
*Lactobacillus* sp./*Enterococcus* sp.	8.99 ^a^	8.77 ^b^	8.77 ^b^	8.97 ^a^	0.18	8.89	8.87	8.88	8.88	0.588	0.909	<0.001
*Clostridium leptum* subgroup	8.90	8.80	8.64	8.62	0.17	8.77	8.71	8.85 ^a^	8.63 ^b^	0.129	<0.001	0.318
*Clostridium coccoides*-*Eubacterium rectale* cluster	9.14	8.98	8.80	8.73	0.16	8.97 ^a^	8.85 ^b^	9.06 ^a^	8.77 ^b^	0.002	<0.001	0.198

^a–c^ Means not sharing a common superscript differ significantly (*p* < 0.05); ^1^ control diet with no additives; ^2^ monensin addition (100 ppm); ^3^ nisin preparation (2700 IU/kg diet); ^4^ a mixture of monensin (100 ppm) and nisin (2700 IU/kg diet); ^5^ root-mean-square error; means represent 10 pens of one chick each (10 replicates).

**Table 5 animals-10-00101-t005:** The effect of dietary supplementation of nisin alone or in combination with monensin on selected microbial populations (log CFU/g digesta) in the cecal content determined by DAPI staining and fluorescent in situ hybridization (FISH), Experiment 1.

Item	Treatments				RMSE ^5^	Main Effects				*p*-Value		
	NA ^1^	MON ^2^	NIS ^3^	MON + NIS ^4^		MON		NIS		Effect of Treatments		Interaction
						−	+	−	+	MON	NIS	MON × NIS
DAPI	11.01	11.07	11.06	11.17	0.14	11.0 ^b^	11.11 ^a^	11.04 ^b^	11.12 ^a^	0.012	0.011	0.468
Enterobacteriaceae	10.48 ^a^	9.52 ^c^	9.63 ^b^^,c^	9.73 ^b^	0.28	10.08 ^a^	9.61 ^b^	9.94 ^a^	9.68 ^b^	<0.001	<0.001	<0.001
*Bacteroides*-*Prevotella* cluster	10.44 ^a^	9.86 ^b^	9.78 ^b^	9.79 ^b^	0.23	10.14 ^a^	9.83 ^b^	10.11 ^a^	9.79 ^b^	<0.001	<0.001	<0.001
*Clostridium perfringens*	10.68 ^a^	9.68 ^c^	9.66 ^c^	9.82 ^b^	0.18	10.21 ^a^	9.75 ^b^	10.13 ^a^	9.75 ^b^	<0.001	<0.001	<0.001
*Lactobacillus* sp./*Enterococcus* sp.	10.73 ^a^	9.82 ^c^	9.88 ^c^	10.11 ^b^	0.13	10.34 ^a^	9.95 ^b^	10.22 ^a^	10.01 ^b^	<0.001	<0.001	<0.001
*Clostridium leptum* subgroup	10.46 ^a^	9.83 ^b,c^	9.71 ^b^	9.86 ^b^	0.23	10.11 ^a^	9.84 ^b^	10.11 ^a^	9.80 ^b^	<0.001	<0.001	<0.001
*Clostridium coccoides*-*Eubacterium rectale* cluster	10.57 ^a^	10.23 ^b^	10.05 ^c^	10.2 ^b^	0.18	10.33 ^a^	10.23 ^b^	10.38 ^a^	10.14 ^b^	0.010	<0.001	<0.001

^a–c^ Means not sharing a common superscript differ significantly (*p* < 0.05); ^1^ control diet with no additives; ^2^ monensin addition (100 ppm); ^3^ nisin preparation (2700 IU/kg diet); ^4^ a mixture of monensin (100 ppm) and nisin (2700 IU/kg diet); ^5^ root-mean-square error; means represent 10 pens of one chick each (10 replicates).

**Table 6 animals-10-00101-t006:** The effect of dietary supplementation of nisin alone or in combination with monensin on organic acid concentrations in the jejunal content (µmol/g), Experiment 1.

Item	Treatments				RMSE ^5^	Main Effects				*p*-Value		
	NA ^1^	MON ^2^	NIS ^3^	MON + NIS ^4^		MON		NIS		Effects of Treatments		Interaction
						−	+	−	+	MON	NIS	MON × NIS
Acetic acid	2.13	1.86	1.62	1.71	0.43	1.86	1.79	1.99 ^a^	1.67 ^b^	0.537	0.024	0.211
Propionic acid	0.12	0.05	0.02	0.03	0.11	0.07	0.04	0.09	0.03	0.358	0.097	0.339
Iso-butyric acid	0.00	0.04	0.00	<0.01	0.04	0.00	0.02	0.02	<0.01	0.104	0.153	0.177
Butyric acid	0.03	0.19	0.03	0.03	0.25	0.03	0.11	0.11	0.03	0.358	0.302	0.330
Iso-valeric acid	0.54	0.09	0.08	0.09	0.69	0.30	0.09	0.30	0.08	0.333	0.326	0.296
Valeric acid	0.01	0.01	<0.01	0.02	0.03	0.01	0.02	0.01	0.01	0.332	0.912	0.422
Sum of VFA ^6^	2.83	2.24	1.76	1.87	0.78	2.27	2.06	2.52 ^a^	1.81 ^b^	0.363	0.007	0.164
Profile C2 ^7^, %	83.47	85.19	92.18	91.32	12.99	88.05	88.26	84.38	91.75	0.925	0.085	0.759
Profile C3 ^8^, %	4.96	2.71	1.41	1.44	4.62	3.09	1.97	3.77	1.34	0.435	0.108	0.479
Profile C4 ^9^, %	1.12	5.95	1.69	0.41	7.24	1.42	3.70	3.66	1.57	0.345	0.372	0.281

^a,b^ Means not sharing a common superscript differ significantly (*p* < 0.05); ^1^ control diet with no additives; ^2^ monensin addition (100 ppm); ^3^ nisin preparation (2700 IU/kg diet); ^4^ a mixture of monensin (100 ppm) and nisin (2700 IU/kg diet); ^5^ root-mean-square error; ^6^ volatile fatty acids; ^7^ acetic acid profile; ^8^ propionic acid profile; ^9^ butyric acid profile; means represent 10 pens of one chick each (10 replicates).

**Table 7 animals-10-00101-t007:** The effect of dietary supplementation of nisin alone or in combination with monensin on organic acid concentrations in the cecal content (µmol/g), Experiment 1.

Item	Treatments				RMSE ^5^	Main Effects				*p*-Value		
	NA ^1^	MON ^2^	NIS ^3^	MON + NIS ^4^		MON		NIS		Effects of Treatments		Interaction
						−	+	−	+	MON	NIS	MON × NIS
Acetic acid	45.34	62.11	66.68	66.64	19.29	56.58	64.37	54.17 ^b^	66.66 ^a^	0.197	0.050	0.183
Propionic acid	4.15	4.60	5.60	4.45	2.32	4.92	4.52	4.39	5.02	0.615	0.397	0.288
Iso-butyric acid	0.46	0.54	0.95	0.37	0.55	0.72	0.46	0.50	0.66	0.155	0.373	0.065
Butyric acid	10.05	15.49	14.72	18.09	6.023	12.51 ^b^	16.79 ^a^	12.91	16.40	0.030	0.079	0.595
Iso-valeric acid	0.74	0.67	0.78	0.60	0.18	0.77 ^a^	0.64 ^b^	0.70	0.70	0.050	0.977	0.367
Valeric acid	0.94	1.05	1.07	1.02	0.24	1.02	1.04	1.01	1.05	0.720	0.587	0.321
Sum of VFA ^6^	61.30	84.46	89.81	91.00	26.90	76.31	87.73	73.49	90.41	0.177	0.058	0.211
Profile C2 ^7^, %	57.67	73.93	74.21	66.16	19.73	66.38	70.05	66.23	70.18	0.554	0.536	0.063
Profile C3 ^8^, %	5.33	5.55	6.35	4.32	2.33	5.86	4.94	5.45	5.33	0.221	0.881	0.141
Profile C4 ^9^, %	12.54	17.80	16.33	17.69	5.25	14.54	17.74	15.31	17.01	0.061	0.317	0.254

^a,b^ Means not sharing a common superscript differ significantly (*p* < 0.05); ^1^ control diet with no additives; ^2^ monensin addition (100 ppm); ^3^ nisin preparation (2700 IU/kg diet); ^4^ a mixture of monensin (100 ppm) and nisin (2700 IU/kg diet); ^5^ root-mean-square error; ^6^ volatile fatty acids; ^7^ acetic acid profile; ^8^ propionic acid profile; ^9^ butyric acid profile; means represent 10 pens of one chick each (10 replicates).

**Table 8 animals-10-00101-t008:** The effect of dietary supplementation of nisin alone or in combination with monensin on ileal histomorphology (µm) of broiler chickens, Experiment 1.

Item	Treatments				RMSE ^5^	Main Effects				*p*-Value		
	NA ^1^	MON ^2^	NIS ^3^	MON + NIS ^4^		MON		NIS		Effects of Treatments		Interaction
						−	+	2212	+	MON	NIS	MON × NIS
Villus high	1084	1099	1045	1027	129.1	1064	1063	1092	1036	0.957	0.187	0.693
Crypt depth	108	112	109	105	13.5	109	109	110	107	0.976	0.491	0.429
V:C ratio	10.2	10.0	9.7	9.9	1.7	10.0	10.0	10.0	9.8	0.979	0.613	0.652
Mucosa thickness	1198	1218	1161	1140	130.8	1179	1179	1208	1151	0.974	0.179	0.629
Muscular layer thickness	164	165	162	159	26.1	1639	162	165	160	0.842	0.619	0.825

^1^ control diet with no additives; ^2^ monensin addition (100 ppm); ^3^ nisin preparation (2700 IU/kg diet); ^4^ a mixture of monensin (100 ppm) and nisin (2700 IU/kg diet); ^5^ root-mean-square error; means represent 10 pens of one chick each (10 replicates).

**Table 9 animals-10-00101-t009:** The effect of dietary supplementation of nisin on the growth performance of broiler chickens, Experiment 2.

Item	Treatment					RMSE ^6^	*p*-Value
	NA ^1^	NIS_100_ ^2^	NIS_200_ ^3^	NIS_400_ ^4^	NIS_800_ ^5^		
BWG ^7^, g							
1–14 d	366	368	366	369	379	2.4	0.382
14–35 d	1640	1704	1745	1655	1690	12.4	0.056
1–35 d	2006	2072	2111	2024	2069	13.5	0.097
FI ^8^, g							
1–14 d	540	541	547	523	539	3.2	0.191
14–35 d	2719	2729	2711	2695	2729	16.1	0.964
1–35 d	3259	3269	3258	3218	3268	16.9	0.877
FCR ^9^, g:g							
1–14 d	1.48	1.47	1.49	1.42	1.43	0.01	0.159
14–35 d	1.66 ^a^	1.60 ^b^	1.55 ^c^	1.63 ^ab^	1.62 ^b^	0.01	<0.001
1–35 d	1.63 ^a^	1.58 ^c^	1.54 ^b^	1.59 ^b^	1.58 ^b^	0.01	<0.001

^a–c^ Means not sharing a common superscript differ significantly (*p* < 0.05); ^1^ control diet with no additives; ^2^ diet with the addition of nisin preparation (100 IU/kg diet); ^3^ nisin supplementation (200 IU/kg diet); ^4^ nisin preparation (400 IU/kg diet); ^5^ nisin addition (800 IU/kg diet); ^6^ root-mean-square error; ^7^ body weight gain; ^8^ feed intake; ^9^ feed conversion ratio; means represent 10 pens of 10 chicks each.

**Table 10 animals-10-00101-t010:** The effect of dietary supplementation of nisin on the selected internal organs of broiler chickens, Experiment 2.

Item	Treatment					RMSE ^6^	*p*-Value
	NA ^1^	NIS_100_ ^2^	NIS_200_ ^3^	NIS_400_ ^4^	NIS_800_ ^5^		
Bursa of Fabricius	0.18	0.15	0.17	0.17	0.17	0.04	0.523
Spleen	0.10	0.12	0.12	0.14	0.13	0.04	0.193
Pancreas	0.28	0.27	0.25	0.25	0.27	0.04	0.214
Liver	2.83	2.85	2.95	2.88	2.90	0.29	0.820

^1^ control diet with no additives; ^2^ diet with the addition of nisin preparation (100 IU/kg diet); ^3^ nisin supplementation (200 IU/kg diet); ^4^ nisin preparation (400 IU/kg diet); ^5^ nisin addition (800 IU/kg diet); ^6^ root-mean-square error; means represent 10 pens of one chicks each (10 replicates).

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
