# Peer review of "Nisin as a Novel Feed Additive: The Effects on Gut Microbial Modulation and Activity, Histological Parameters, and Growth Performance of Broiler Chickens"

_animals, 2020, doi:10.3390/ani10010101_

Round 1

Reviewer 1 Report

All results are lebeled as Experiment 1, please check.

There are some typos and grammatical errors that require a language check.

Conclusions are bit of a stretch that nisin has a similar mode of action as monensin when the actual mode of action of monensin (anti-coccidial) wasn't even explored. Please rewrite the conclusion and interpretation to remove these kind of generic claims.

Author Response

Response to reviewers

We thank the Editor and Referees for their interest in our work and for their helpful comments which will greatly improve the manuscript. We have done our best to respond to all the points raised in the reviews. The Referees have raised certain points and we appreciate the opportunity to clarify our research objectives and results. As indicated below, we have checked all the general and specific comments pointed out line by line by the Referees and have made the necessary changes accordingly to their indications.

Reviewer 1:

Comment 1: All results are lebeled as Experiment 1, please check.

Answer: The manuscript was double-checked in these terms and corrected.

Comment 2: There are some typos and grammatical errors that require a language check.

Answer: The manuscript was checked by the native speaker. The Editing Certificate from American Journal Experts (www.aje.com), i.e., recommended by the following publishers - Nature, Springer, Cambridge University Press, Elsevier, etc. was uploaded with the other files.

Comment 3: Conclusions are bit of a stretch that nisin has a similar mode of action as monensin when the actual mode of action of monensin (anti-coccidial) wasn't even explored. Please rewrite the conclusion and interpretation to remove these kind of generic claims.

Answer: The Authors want to thank Reviewer for this comment. It is well-known that monensin is commonly used ionophore coccidiostat in poultry production, however, the antimicrobial properties of this compound are also well-studied. In the present study, the only effect on the selected microbial populations was determined. The anti-coccidial effect was not included in the aim of the study, as the Reviewer point out rightly. In the available literature the ionophore coccidiostats, i.e., monensin, as well as in particular salinomycin used to be used as an antimicrobial agent to compare their effect with other proposed substances. A few examples are listed below:

1) KieroÅ„czyk, B., PruszyÅ„ska-OszmaÅ‚ek, E., ÅšwiÄ…tkiewicz, S., Rawski, M., DÅ‚ugosz, J., Engberg, R. M., & Józefiak, D. (2016). The nisin improves broiler chicken growth performance and interacts with salinomycin in terms of gastrointestinal tract microbiota composition. Journal of Animal and Feed Sciences25(4), 309-316.

2) Józefiak, D., Sip, A., Rawski, M., Steiner, T., & Rutkowski, A. (2011). The dose response effects of liquid and lyophilized Carnobacterium divergens AS7 bacteriocin on the nutrient retention and performance of broiler chickens. Journal of Animal and Feed Sciences20(3), 401-411.

3) Józefiak, D., Sip, A., Rawski, M., Rutkowski, A., Kaczmarek, S., Hojberg, O., ... & Engberg, R. M. (2011). Dietary divercin modifies gastrointestinal microbiota and improves growth performance in broiler chickens. British poultry science52(4), 492-499.

4) Józefiak, D., Sip, A., Kaczmarek, S., & Rutkowski, A. (2010). The effects of Carnobacterium divergens AS7 bacteriocin on gastrointestinal microflora in vitro and on nutrient retention in broiler chickens. Journal of Animal and Feed Sciences19(3), 460-467.

From this reason, the Authors did not claim that nisin as a novel growth promoter can fully replace monensin in terms of the anti-coccidial agent but as an antimicrobial factor. The following Conclusions section is suggested by the Authors:

The results of the current experiments emphasized that nisin addition to broiler chicken diets may be considered a novel and natural growth promoter that improves feed utilization even at low levels. Moreover, it was confirmed that nisin plays a significant role in the positive modulation of the microecosystem of the GIT of broiler chickens via the reduction in the proliferation and populations of potentially pathogenic bacteria, which may negatively alter nutrient utilization. A limitation of bacterial activity was observed in the jejunal digesta, while cecal fermentation tended to enhance. Additionally, a positive interaction between bacteriocin nisin and ionophore coccidiostat – monensin was observed without any additive or synergetic effect. The findings of the present study suggest that nisin exerts a mode of action similar to that of monensin in the scope of antimicrobial properties. Additional studies in terms of the evaluation of anticoccidial properties of nisin are recommended.”

Kind regards,

Reviewer 2 Report

The current manuscript evaluates the effects of nisin as a novel feed additive for broilers by characterizing the gut microbial modulation and activity, the histological parameters and the growth performance. The topic of the research is interesting and may provide novel, useful information for the scientific community. The manuscript is overall well-written and clear, but there are some minor aspects that need to be addressed in order to make it suitable for publication in the Animals journal.

1) Simple Summary and Abstract: I have no comments.

2) Introduction: I have no comments.

3) Materials and Methods:

Diets: the sum of the ingredients for both the diets (Table 1) is slightly higher than 1000 (1000,2 and 1000,1). Please, check if there are small errors in the ingredient amounts or if the formulation is ok. Lines 150-152: how were the formalin-fixed ileal samples processed, cut and stained? Please, give more information. Lines 152-155: why were the organs not submitted to histopathological examination? It could have been an added value in terms of animal health assessment. Histological analysis: how many villi per serial slide did you measure? Even if this information is reported in the provided reference, I suggest to recall it in the text. You measured the crypt depth and the villus height to crypt depth ratio too (as you reported the corresponding results in the Table 8), but you did not herein specified the related methods. Please, add this information (as well as the number of crypts you measured). How many times did you measure the mucosal thickness and the muscolaris per sample? Same consideration alredy made for the number of villi: even if this information is reported in the provided reference, I suggest to recall it in the text.

4) Results:

Line 222: the p value is missing; Lines 257-259: if the p values are not significant, please report them as "p > 0.05" (as you did in the previous sections).

5) Discussion:

Lines 303-305 and 311-312: Lactobacillus spp is generally considered a positive genus for its well-recognized and documented probiotic properties. Therefore, I would not be so drastic in considering its reduction a positive finding. I suggest to extend the discussion about the SCFA findings, trying to give more information about the different types you measured and, most of all, to relate them to the microbial populations that changed. For example, you observed an increase in butyric and iso-valeric acids in monensin-supplemented birds, but you did not comment these data. Furthermore, you did not give any potential explanation about the differences between the jejunal and the cecal fermentation. All these information could be an added value for your research work. I suggest to extend also the discussion about the gut histomorphology. It could help the reader giving more information about the practical application of such measurements (i.e., Vh as indicator of digestion and absorption capacity of gut, Cd as indicator of enterocyte turnover, etc..).

6) Conclusions: I have no comments.

Author Response

Response to reviewers

We thank the Editor and Referees for their interest in our work and for their helpful comments which will greatly improve the manuscript. We have done our best to respond to all the points raised in the reviews. The Referees have raised certain points and we appreciate the opportunity to clarify our research objectives and results. As indicated below, we have checked all the general and specific comments pointed out line by line by the Referees and have made the necessary changes accordingly to their indications.

Reviewer 2:

The current manuscript evaluates the effects of nisin as a novel feed additive for broilers by characterizing the gut microbial modulation and activity, the histological parameters and the growth performance. The topic of the research is interesting and may provide novel, useful information for the scientific community. The manuscript is overall well-written and clear, but there are some minor aspects that need to be addressed in order to make it suitable for publication in the Animals journal.

Answer: The Authors want to thank the Reviewer for the pleasant description.

Simple Summary and Abstract: I have no comments. Introduction: I have no comments. Materials and Methods:

Diets: the sum of the ingredients for both the diets (Table 1) is slightly higher than 1000 (1000,2 and 1000,1). Please, check if there are small errors in the ingredient amounts or if the formulation is ok.

Answer: The Authors fully agree with the Reviewer. It is a result of rounding out the ingredients parts. Table 1 was checked and corrected.

Lines 150-152: how were the formalin-fixed ileal samples processed, cut and stained? Please, give more information.

Answer: The whole procedure used to analyse the histomorphological parameters was based on Rawski et al., 2016. The section titled ‘Histological Analysis’ was extended as follow:

The histological analysis of ileal samples was performed according to Rawski et al. [31]. Briefly, the ileal samples were fixed immediately in freshly prepared formaldehyde solution (40 g/L of formaldehyde prepared in 0.01 M PBS, pH=7.4) and incubated 12 h. Ileal fragments were dehydrated in alcohol dilutions, stowed in xylene, and embedded in paraffin. At the least 10 slides (5 μm) per block were stained using haematoxylin and eosin. The obtained material was analyzed under a light microscope (Axiophot, Carl Zeiss, Germany) with 5 x 5 magnification. The length of villi was measured from the top of the epithelium villi to the junction with the crypt. In the cross-sections, the lengths of all villi with a complete structure were measured. Destroyed villi were excluded from the trial. Mucosal thickness was determined as the distance between the mucosal epithelium and the muscular layer, and the muscularis was determined as the inner circular and outer longitudinal layers of smooth-muscle cells [38]. The measurements were made on 10 serial slides using a micrometer glass master (0.01 mm, PZO, Warsaw, Poland) and treated as the means.’

Lines 152-155: why were the organs not submitted to histopathological examination?
It could have been an added value in terms of animal health assessment. Histological analysis: how many villi per serial slide did you measure? Even if this information is reported in the provided reference, I suggest to recall it in the text. You measured the crypt depth and the villus height to crypt depth ratio too (as you reported the corresponding results in the Table 8), but you did not herein specified the related methods. Please, add this information (as well as the number of crypts you measured). How many times did you measure the mucosal thickness and the muscolaris per sample? Same consideration alredy made for the number of villi: even if this information is reported in the provided reference, I suggest to recall it in the text.

Answer: Please, find the above-mentioned information. The section titled ‘Histological Analysis’ was extended.

Results:

Line 222: the p value is missing;

Answer: The missing p-value was corrected.

Lines 257-259: if the p values are not significant, please report them as "p > 0.05" (as you did in the previous sections).

Answer: The paragraph has been corrected as follow:

Table 8 presents the results of the ileal histomorphometry measurements. No significant interaction (p > 0.05), as well as effects of experimental factors (p > 0.05) were noticed in the case of villus high, crypt depth, their ratio, mucosa, and muscular layer thickness.’

Discussion:

Lines 303-305 and 311-312: Lactobacillus spp is generally considered a positive genus for its well-recognized and documented probiotic properties. Therefore, I would not be so drastic in considering its reduction a positive finding.

Answer: The Authors agree with the Reviewer that the Lactobacilli group are mainly considered as a probiotic microbiota. However, from the Authors point of view in the animal production where the high performance is crucial to its profitability, the possible reduction of coefficients of ether extract digestibility, and simultaneously the limitation of energy availability for birds via deconjugation of bile sides by the Lactobacilli group is the negative effect. The following article was cited in the text:

Klaver, F.A.; Van der Meer, R. The assumed assimilation of cholesterol by Lactobacilli and Bifidobacteriumbifidum is due to their bile salt-deconjugating activity. Appl. Environ. Microbiol. 1993, 59, 1120–1124.

‘The positive effects of nisin application in broiler chicken diets on changes in the microbiota consist of not only the reduction in pathogen occurrence in the chicken GIT but also lowering the competition for nutrients between bacteria and the host, improved energy utilization through decreasing the number of bacteria from the genera Lactobacillus, Clostridium, as well as Bacteroides [46,47].’

I suggest to extend the discussion about the SCFA findings, trying to give more information about the different types you measured and, most of all, to relate them to the microbial populations that changed. For example, you observed an increase in butyric and iso-valeric acids in monensin-supplemented birds, but you did not comment these data.

Answer: Authors added information about above-mentioned results in the Discussion section.

‘Additionally, the effect of monensin was noticed by the positive reduction of iso-valeric acid concentration, which is a component of putrefactive SCFA (PSFA). It is well known that PSCFA is related to protein fermentation in the ceca by, e.g., Clostridium perfringens, and Enterobacteriaceae [53,54]. It is in agreement with obtained results, where the proliferation of abovementioned bacteria was inhibited. Moreover, the decreasing number of volatile fatty acid-producing strains in the ceca result in increased concentrations of propionic, acetic, as well as butyric acid [55]. In the current study, only butyric acid fermentation has been enhanced by monensin addition, while the Clostridium leptum subgroup and Clostridium coccoides - Eubacterium rectale cluster were lowered by the coccidiostat. Nevertheless, the increased level of their activity may have a beneficial impact on the GIT microbiota populations [56].’

The additional citations were added to the manuscript:

Terada, A.; Hara, H.; Sakamoto, J.; Sato, N.; Takagi, S.; Mitsuoka, T.; Mino, R.; Hara, K.; Fujimori, I.; Yamada, T. Effects of dietary supplementation with lactosucrose (4G-β-D-Galactosylsucrose) on cecal flora, cecal metabolites, and performance in broiler chickens. Poult. Sci. 1994, 73, 1663–1672. ZduÅ„czyk, Z.; Jankowski, J.; Rutkowski, A.; Sosnowska, E.; Drażbo, A.; ZduÅ„czyk, P.; JuÅ›kiewicz, J. The composition and enzymatic activity of gut microbiota in laying hens fed diets supplemented with blue lupine seeds. Anim. Feed Sci. Technol. 2014, 191, 57–66. Corrier, D.E.; Hinton Jr, A.; Ziprin, R.L.; Beier, R.C.; DeLoach, J.R. Effect of dietary lactose on cecal pH, bacteriostatic volatile fatty acids, and Salmonella typhimurium colonization of broiler chicks. Avian Dis.1990, 617–625. Leeson, S.; Namkung, H.; Antongiovanni, M.; Lee, E.H. Effect of butyric acid on the performance and carcass yield of broiler chickens. Poult. Sci. 2005, 84, 1418–1422.

Furthermore, you did not give any potential explanation about the differences between the jejunal and the cecal fermentation. All these information could be an added value for your research work.

Answer: The Authors appreciate the Reviewer suggestion, and decided to add following explanation:

‘In the present study, the effects of nisin on the microbiota fermentation (ileal and cecal) may be explained by its main antimicrobial targeting. It is well documented that bacteriocins inhibit the growth and development of bacteria especially in the case of closely related taxa, i.e., across genera or the same species [48]. Due to this fact, the nisin produced by the L. lactis subsp. lactis may have the main impact on lactic acid bacteria (LAB), thus the microbial fermentation is reduced in the higher GIT segments where they occur as dominant, i.e., the crop (109 cells g -1), gizzard (108 cells g -1) as well as small intestine (109-11 g cells g -1) [49–51]. Contrary to the ceca where the LAB populations (Bacillus-Lactobacillus-Streptococcus subdivision) are in minority [52]. The abovementioned mechanism seems to be confirmed by the fact that the fermentation in ceca was not reduced as much as in the upper parts due to the presence of wide spectra of bacterial populations which could be resistant to nisin activity.’

Additional citations were used:

Cotter, P.D. C. Hill ad RP Ross, 2005. Bacteriocins: Developing innate immunity for food. Nat. Rev. Microbiol 3, 777–788. KieroÅ„czyk, B.; Rawski, M.; DÅ‚ugosz, J.; ÅšwiÄ…tkiewicz, S.; Józefiak, D. Avian crop function–a review. Ann. Anim. Sci. 2016, 16, 653–678. Yeoman, C.J.; Chia, N.; Jeraldo, P.; Sipos, M.; Goldenfeld, N.D.; White, B.A. The microbiome of the chicken gastrointestinal tract. Anim. Heal. Res. Rev. 2012, 13, 89–99. Oakley, B.B.; Lillehoj, H.S.; Kogut, M.H.; Kim, W.K.; Maurer, J.J.; Pedroso, A.; Lee, M.D.; Collett, S.R.; Johnson, T.J.; Cox, N.A. The chicken gastrointestinal microbiome. FEMS Microbiol. Lett. 2014, 360, 100–112.

I suggest to extend also the discussion about the gut histomorphology. It could help the reader giving more information about the practical application of such measurements (i.e., Vh as indicator of digestion and absorption capacity of gut, Cd as indicator of enterocyte turnover, etc..).

Answer: The Authors would like to thank for the Reviewer suggestion. However, no significant results in this term were obtained in the current study. In the Authors opinion, this is enough informative, due to the possibility of future registration of nisin as a feed additive. No detrimental effect on the GIT histomorphology is a positive effect. It should be also emphasized that the current manuscript is a continuation of a previous trial described in Kierończyk et al. (2017), in which nisin and monensin were added to broiler chicken diets and the effect on growth performance parameters, coefficients of apparent digestibility of nutrients, tibiotarsus chemical composition, and length and weight of selected sections of the gastrointestinal tract was analyzed. From the Authors point of view, these results give wide spectra of traits which are informative for future readers. Additionally, the Authors presented results from other papers which confirmed no effect on the GIT histomorphology.

Conclusions: I have no comments.

Kind regards,

Reviewer 3 Report

The aim of the study was to determine the effect of nisin (baceriocin produced by most known bactria) on the composition and activity of microbiota, as well as histomorphology and production results of broiler chickens. The results obtained are important for the feed industry, producers of broiler chickens and consumers of chicken meat.  Before printing, the manuscript requires additions and a minor correction. The list of proposed changes is below.

General comments:

All paper: broiler chickens instead of broilers, there are broilers of other species of poultry

All paper: microbiota instead of microecology

Correction of page numbers from page 6, repetition of page numbers

Specific comments:

L 4 broiler chickens instead of broilers

L37 caeca or ceca? British or American sperling, please check the authors' instructions

L36 affected the microbiota instead of the current form

L40 broiler chickens instead of broilers

L72 broiler chickens instead of broilers

L74 EFSA (European Food Safety Authority) instead of the current form

L94 slaughtered and eviscerated, instead of dissection – a dissection is cutting the carcase. No data on the composition of the carcass in this paper

L 95 caeca or ceca?

L101 determined body weight and feed intake, BWG (body weight gain) and FCR calculated,

 please make a correction. Add information about the type of scales, name, manufacturer's data, measurement accuracy that was recorded FI and BW

L106 + please add information about the type of building (closed, no windows?), Temperature range from-to, humidity range from-to, not exceeding the maximum concentration of CO2 (3000 ppm) and NH3 (20 ppm), length of the light day, light intensity, type of light (and were they in accordance with the instructions for broiler chickens Ross 308?), type of floor (straw litter?)

Table 1 - total ingredient starter diet must be 1000 g, but now  is 1000.2, while grower / diet finisher is 1000.1 and must be 1000 g in 1 kg of feed mixture

In table 1 enter the CP content for “Rapeseed meal”, crude fiber or crude fibre? Check

L144 using a pH meter?

L154 “after slaughter” instead of after dissection

L 154-155 using the electronic balance PS 600 /C/2 (Radwag, Radom, Poland). The measurement was made with accuracy to ........ "instead of the current form

L197, 214, 231 a-c instead of a-b

L205-207 please check the compliance of the description with the data in Table 4, no description for the characteristic "Clostridium leptum subgroup"

L 222 (p < 0.001) instead of (p =)

L218-220 please delete the sentence "A significant ... ..p = 0.468" is repeated on lines 224-226

L242 add "in caeca" after (p = 0.177)

L257-259 The description does not match the data in Table 8

L262, 283  – remove “a-b Means not sharing….”, because there are no significant differences

Author Response

Response to reviewers

We thank the Editor and Referees for their interest in our work and for their helpful comments which will greatly improve the manuscript. We have done our best to respond to all the points raised in the reviews. The Referees have raised certain points and we appreciate the opportunity to clarify our research objectives and results. As indicated below, we have checked all the general and specific comments pointed out line by line by the Referees and have made the necessary changes accordingly to their indications.

Reviewer 3

The aim of the study was to determine the effect of nisin (baceriocin produced by most known bacteria) on the composition and activity of microbiota, as well as histomorphology and production results of broiler chickens. The results obtained are important for the feed industry, producers of broiler chickens and consumers of chicken meat.  Before printing, the manuscript requires additions and a minor correction. The list of proposed changes is below.

General comments:

All paper: broiler chickens instead of broilers, there are broilers of other species of poultry

Answer: The manuscript has been corrected.

All paper: microbiota instead of microecology

Answer: The manuscript has been corrected.

Correction of page numbers from page 6, repetition of page numbers

Answer: The page numbers have been corrected.

Specific comments:

L4 broiler chickens instead of broilers

Answer: corrected

L37 caeca or ceca? British or American sperling, please check the authors' instructions

Answer: In the Instructions for Authors there is no annotation about British or American English recommendation. The Authors used American English, and the manuscript is unified. Due to this fact, the term 'ceca' is used. According to the Journal recommendation, the manuscript was professionally edited and read by a native English-Speaker (American Journal Expert – recommended by following publishers, Nature, Elsevier, Springer, Cambridge University Press).

L36 affected the microbiota instead of the current form

Answer: corrected

L40 broiler chickens instead of broilers

Answer: corrected

L72 broiler chickens instead of broilers

Answer: corrected

L74 EFSA (European Food Safety Authority) instead of the current form

Answer: The Authors explain the shortcut in line 59, in this case there is no need to expand abbreviation once again.

L94 slaughtered and eviscerated, instead of dissection – a dissection is cutting the carcase. No data on the composition of the carcass in this paper

Answer: The Authors fully agree with the Reviewer. The manuscript has been corrected.

L 95 caeca or ceca?

Answer: in the Instructions for Authors there is no annotation about British or American English recommendation. The Authors used American English, and the manuscript is unified. Due to this fact, the term 'ceca' is used. According to the Journal recommendation, the manuscript was professionally edited and read by a native English-Speaker (American Journal Expert).

L101 determined body weight and feed intake, BWG (body weight gain) and FCR calculated, please make a correction. Add information about the type of scales, name, manufacturer's data, measurement accuracy that was recorded FI and BW

Answer: The Authors made the following corrections:

‘The experiment lasted 35 d, and the growth performance parameters, i.e., body weight and feed intake (FI), were determined, and the body weight gain (BWG) and feed conversion ratio (FCR) was calculated. The abovementioned variables were obtained at 14 and 35 d of age using analytical scale (NVL5101, OHAUS, Switzerland) with accuracy ± 1 g.’

L106 + please add information about the type of building (closed, no windows?), Temperature range from-to, humidity range from-to, not exceeding the maximum concentration of CO2 (3000 ppm) and NH3 (20 ppm), length of the light day, light intensity, type of light (and were they in accordance with the instructions for broiler chickens Ross 308?), type of floor (straw litter?)

Answer: The Authors added the following phrase:

The housing conditions were the same in both experiments, and they were as follows: the birds were kept in floor pens (1.00 x 1.00 m; straw litter) over 35 d. Stock density was established at 10 birds per square meter. Additionally, 9,000 birds of the same origin (hatchery) were used to imitate the commercial environment production conditions. The closed chicken house was equipped with an artificial light program (fluorescent lights), automatic heaters, and forced ventilation. The birds were given 23 h of light and 1 h of dark for the first week, followed by 19 h of light and 5 h without from 7 to 21 d of age. From 22-35 d of age, the lighting system was similar to that of the first week. The light intensity was set at 20 lx according to the EU directive (2007/43/EC). The temperature inside the building was set up at 32-33°C at the beginning of the test and was reduced by 2-3°C each week. On the 28th day, the temperature was set at 21°C and was approximately 18°C at the end of the experiment. The humidity level was in the range of 50-60%. During the experimentation, the maximum concentration of CO2, as well as NH3 did not exceed 3000 ppm, and 10 ppm, respectively. The rearing conditions were set up accordingly to AVIAGEN guidelines.

Table 1 - total ingredient starter diet must be 1000 g, but now is 1000.2, while grower / diet finisher is 1000.1 and must be 1000 g in 1 kg of feed mixture

Answer: The Authors fully agree with the Reviewer. It is a result of rounding out the ingredients parts. Table 1 was checked and corrected.

In table 1 enter the CP content for “Rapeseed meal”, crude fiber or crude fibre? Check

Answer: The CP content for rapeseed meal was added, i.e. 34%.
As was mentioned above, the Authors used American English spelling, due to this fact ‘fiber’ is correct.

L144 using a pH meter?

Answer: corrected

L154 “after slaughter” instead of after dissection

Answer: corrected

L 154-155 using the electronic balance PS 600 /C/2 (Radwag, Radom, Poland). The measurement was made with accuracy to ........ "instead of the current form

Answer: corrected

L197, 214, 231 a-c instead of a-b

Answer: corrected

L205-207 please check the compliance of the description with the data in Table 4, no description for the characteristic “Clostridium leptum subgroup”

Answer: The Clostridium leptum subgroup was described in lines 220-221.

Furthermore, nisin supplementation significantly decreased the number of the Clostridium leptum subgroup (p < 0.001) and the Clostridium coccoides – Eubacterium rectale cluster (p < 0.001).’

However, the Authors did notice that superscripts which should show significance between groups had been mentioned simultaneously when the interaction was not significant. Due to this Authors have made a correction of all tables in the manuscript and deleted not necessary superscripts which could be make misunderstood of future Readers.

L 222 (p < 0.001) instead of (p =)

Answer: corrected

L218-220 please delete the sentence "A significant ... ..p = 0.468" is repeated on lines 224-226

Answer: corrected

L242 add "in caeca" after (p = 0.177)

Answer: corrected

L257-259 The description does not match the data in Table 8

Answer: The Authors would like to thank the Reviewer who has absolutely right. The paragraph has been corrected as follow:

Table 8 presents the results of the ileal histomorphometry measurements. No significant interaction (p > 0.05), as well as effects of experimental factors (p > 0.05) were noticed in the case of villus high, crypt depth, their ratio, mucosa, and muscular layer thickness.’

L262, 283 – remove “a-b Means not sharing….”, because there are no significant differences

Answer: corrected

Kind regards,
